# Down-regulation of hepatic expression of GHR/STAT5/IGF-1 signaling pathway fosters development and aggressiveness of HCV-related hepatocellular carcinoma: Crosstalk with Snail-1 and type 2 transforming growth factor-beta receptor

**Mona A. Abu El-Makarem**[1]*, **Mariana F. Kamel**[2,3], **Ahmed A. Mohamed**[1], **Hisham A. Ali**[1], **Mahmoud R. Mohamed**[1], **Alaa El-Deen M. Mohamed**[1], **Ahmed M. El-Said**[1], **Mahmoud G. Ameen**[4], **Alshymaa A. Hassnine**[5], **Hatem A. Hassan**[1]

**1** Department of Internal Medicine, School of Medicine, Minia University, Minia, Egypt, **2** Department of Pathology, School of Medicine, Minia University, Minia, Egypt, **3** Department of Pathology, Minia Oncology Center, Minia, Egypt, **4** Department of Pathology, South Egypt Cancer Institute, Assuit University, Assuit, Egypt, **5** Department of Tropical Medicine and Gastroenterology, School of Medicine, Minia University, Minia, Egypt

* mona.makarim@yahoo.com

## Abstract

### Background and aims

So far, few clinical trials are available concerning the role of growth hormone receptor (GHR)/signal transducer and activator of transcription 5 (STAT5)/insulin like growth factor-1 (IGF-1) axis in hepatocarcinogenesis. The aim of this study was to evaluate the hepatic expression of GHR/STAT5/IGF-1 signaling pathway in hepatocellular carcinoma (HCC) patients and to correlate the results with the clinico-pathological features and disease outcome. The interaction between this signaling pathway and some inducers of epithelial-mesenchymal transition (EMT), namely Snail-1 and type 2 transforming growth factor-beta receptor (TGFBR2) was studied too.

### Material and methods

A total of 40 patients with HCV-associated HCC were included in this study. They were compared to 40 patients with HCV-related cirrhosis without HCC, and 20 healthy controls. The hepatic expression of GHR, STAT5, IGF-1, Snail-1 and TGFBR2 proteins were assessed by immunohistochemistry.

### Results

Compared with cirrhotic patients without HCC and healthy controls, cirrhotic patients with HCC had significantly lower hepatic expression of GHR, STAT5, and IGF-1proteins. They also displayed significantly lower hepatic expression of TGFBR2, but higher expression of

**Data Availability Statement:** All relevant data, metadata and methods are included in the paper without limitations.

**Funding:** The authors received no specific funding for this work.

**Competing interests:** The authors have declared that no competing interests exist.

**Abbreviations:** HCC, hepatocellular carcinoma; HCV, hepatitis C virus; IGF-1, insulin like growth factor-1; GH, growth hormone; GHR, growth hormone receptor; IGF-1R, type1 IGF receptor; IGF-2R, type 2 IGF receptor; IR, insulin receptor; IGFBP, insulin growth factor binding protein; ALS, acid-labile subunit; JAK2, Janus Kinase2; STAT, signal transducer and activator of transcription; EMT, epithelial-mesenchymal transition; TME, tumor microenvironment; TGF-β, transforming growth factor-beta; TGFBR2, type 2 TGF-β receptor; IHC, immunohistochemistry; AFP, alpha fetoprotein; MELD, model for end-stage liver disease; TNM, Tumor-Node-Metastasis; SD, standard deviation; IQR, interquartile range; OS, overall survival; HR, hazard ratio; CI, confidence interval; KCs, kupffer cells; HSCs, hepatic stellate cells; LSECs, liver sinusoidal endothelial cells; CD, cluster differentiation; TAM, tumor-associated macrophage.

Snail-1 versus the non-HCC cirrhotic patients and controls. Serum levels of alpha-fetoprotein (AFP) showed significant negative correlations with hepatic expression of GHR (r = -0.31; p = 0.029) and STAT5 (r = -0.29; p = 0.04). Hepatic expression of Snail-1 also showed negative correlations with GHR, STAT5, and IGF-1 expression (r = -0.55, p = 0.02; r = -0.472, p = 0.035, and r = -0.51, p = 0.009, respectively), whereas, hepatic expression of TGFBR2 was correlated positively with the expression of all these proteins (r = 0.47, p = 0.034; 0.49, p = 0.023, and r = 0.57, p<0.001, respectively). Moreover, we reported that decreased expression of GHR was significantly associated with serum AFP level>100 ng/ml (p = 0.048), increased tumor size (p = 0.02), vascular invasion (p = 0.002), and advanced pathological stage (p = 0.01). Similar significant associations were found between down-regulation of STAT5 expression and AFP level > 100 ng/ml (p = 0.006), vascular invasion (p = 0.009), and advanced tumor stage (p = 0.007). Also, attenuated expression of IGF-1 showed a significant association with vascular invasion (p < 0.001). Intriguingly, we detected that lower expression of GHR, STAT5 and IGF-1 were considered independent predictors for worse outcome in HCC.

## Conclusion

Decreased expression of GHR/STAT5/IGF-1 signaling pathway may have a role in development, aggressiveness, and worse outcome of HCV-associated HCC irrespective of the liver functional status. Snail-1 and TGFBR2 as inducers of EMT may be key players. However, large prospective multicenter studies are needed to validate these results.

## Introduction

Hepatocellular carcinoma (HCC) is a global health problem, as it is the sixth most prevalent cancer and the fourth cause of cancer-associated deaths [1]. It develops in 80–90% of patients on a background of liver cirrhosis, irrespective of the underlying etiology [2]. Among the different etiological factors of liver cirrhosis, chronic hepatitis C virus (HCV) infection contributes to the most frequent risk factor for developing HCC in Egypt [3] due to its widespread presence in this locality [4].

HCC is highly heterogeneous; from clinical point of view, ~ 80% of patients are diagnosed with advanced stage which makes palliative therapy requisite, in addition to the diverse etiological risk factors [5], and the high 5-year recurrence rate [6]. As yet, the imaging modalities used for HCC diagnosis and staging are comparatively imprecise [7], moreover, the various histopathological subtypes might contradict accurate diagnosis [8]. With respect to the molecular mechanisms, HCC is associated with multiple genetic and epigenetic alterations that co-operate with the tumor microenvironment to hasten hepatocarcinogenesis, tumor progression and metastasis [9]. As a result of these alterations, several signaling pathways are dysregulated, including growth factors e.g insulin like growth factor (IGF) [10].

The cytokine growth hormone (GH) is secreated in a pulsatile manner by somatotropic cells in the lateral wings of the anterior pituitary gland. It fulfills its effects by direct or indirect means; the direct effect is mediated by binding of GH to its receptors (GHR) on target cells by activating the mitogen activated protein kinase/extracellular signal-regulated kinase pathway, whereas the indirect effect is exerted by the way of its effectors, mainly IGF-1. While the GH is the major regulator of IGF-1 production by liver cells, IGF-1 counteracts GH synthesis by a

negative feedback mechanism. IGF-1 is also synthesized by all target tissues, so it acts as an endocrine and autocrine / paracrine hormone [11].

IGF system includes 2 specific ligands; IGF-1 and IGF-2. Both IGFs play a crucial role in somatic growth and anabolic roles in various tissues and organs. IGF-1 exerts its growth promoting effects of GH during postnatal life, however, IGF-2 is responsible for prenatal and fetal growth, that is GH-independent [12]. IGFs mediate their effects through a group of cell surface receptors including type I IGF receptor (IGF-IR), type II IGF receptor (IGF-2R), insulin receptor (IR), and hybrid receptor (IGF-IR/IR). IGF-IR binds IGF-1 with higher affinity than IGF-2 and insulin, while IGF-2 is the only ligand of IGF-2R [13]. In biological fluids, levels of IGF are adjusted by a group of six IGF binding proteins (IGFBPs) that subjected to proteolysis by IGFBP- specific proteases to release bound IGF by decreasing their affinity. The majority of IGF is bound to IGFBP-3, with acid-labile subunit (ALS) forming ternary complexes that act as transporters of IGF and prolong their half-lives in the circulation [14].

GH/GHR interaction evokes activation of various tyrosine kinases including Janus Kinase 2 (JAK2) protein. Subsequently, the activated GHR-JAK 2 complex triggers phosphorylation of signal transducer and activator of transcription 5A (STAT5A) and STAT5B (both referred to as STAT5), in addition to STAT1 and STAT 3, when STAT 5 expression is low or absent. Activated STAT proteins translocate into the nucleus to initiate different gene expression. STAT5B is aminly expressed in liver cells. Hepatic STAT5B guides the transcription of IGF-1, ALS, suppressors of cytokine signaling 2, components of the cytochrome p 450 detoxifying system, as well as many genes related to glucose and lipid metabolism [15].

Taken into account the potent mitogenic and antiapoptotic role of GH and IGF-1 in all cells of the organism [16], it is not surprising that both can influence cancer risk. Compelling evidences obtained from different types of studies have demonstrated significant associations between raised serum levels of IGF-1 and increased risk of various solid tumors [17–19], including HCC [20–22]. Likewise, epidemiological studies have revealed an increased risk of a variety of cancers in patients with acromegaly [23]. Additionally, a high incidence of colorectal cancer and Hodgkin's lymphoma has also been reported in children treated with pituitary–derived GH [24]. However, the causal role of GH and IGF-1 in oncogenesis is still debated; they might serve as permissible agents [25]. Although the activation of STAT5 system has been considered a risk factor for different human cancers [26–28], its role in HCC is context–dependent as it can act as a tumor promoter and a tumor suppressor [29]. To the best of our knowledge, the clinical studies in this research area have been relatively scarce and had discordant results.

EMT is a developmental regulatory reversible process initiated in cancer cells by which epithelial cells acquire the capability to proliferate, invade, and resist apoptosis. EMT is characterized by down-regulation of epithelial markers such as E-cadherin, Claudin family and specific cytokeratin intermediate filament proteins, combined with up-regulation of mesenchymal markers e,g., N-cadherin, α-smooth muscle actin and vimentin [30]. It is triggered by repression of E-cadherin expression via transcription factors; including zinc finger proteins, (Snail 1/2/3 and Zeb 1/2), and basic helix loop-helix proteins (Twist 1/2) [31]. E-cadherin is a cell membrane protein that binds to β-catenin; one of the components of Wnt signaling pathway which allows epithelial cells to firmly attached. Therefore, reduced expression of E-cadherin resulting in translocation of β-catenin into the nucleus leading to induction of EMT via liberation of many transcription factors [32]. At the metastatic site, the epithelial cells return toward its ancestral condition by a process known as mesenchymal-epithelial transition to return the proliferative state to produce metastatic nodules [33].

Snail-1 is the most important transcription factor responsible for E-cadherin repression by mediating histone modification [34]. Additionally, Snail-1 plays a critical role in cell survival

[35], immune regulation [36] as well as, preservation of cancer stem cell-like properties [37]. Snail expression is under control of many signaling molecules released from tumor microenvironment (TME) such as epidermal growth factor, fibroblast growth factor, transforming growth factor-beta (TGF-β), Notch, Wnt, tumor necrosis factor-α, and cytokines [38].

TGF-β is a multifunctional cytokine that signals through heteromerics of type 1 and type 2 TGF-β receptors (TGFBR1, TGFBR2) which activate either Smad family via Smad 2/3/4 complexes or non-Smad cascades such as PI3K/Akt, p38MAPK, MAPK-ERK and JNK pathways [39]. Activated TGF-β could be tumor suppressive or oncogenic as determined by the context. In tumor cells, the TGF-β signaling is unregulated by various mutations or epigenetic changes, thereby; cells become resistant to the suppressive sequel of TGF-β signaling pathway [40]. Down-expression or mutations of TGFBR2 has been reported in various cancers [41–43], inclusive of HCC [44], however, the underlying mechanism has not been clarified yet.

The purpose of this study was to investigate the tissue expression of GHR/STAT5/IGF-1 signaling pathway by immunohistochemistry (IHC) in HCV–associated HCC patients and to correlate the results with the clinico-pathological features and disease outcome. The interplay between this signaling pathway and both Snail-1 and TGFBR2 as inducers of EMT, was evaluated as well.

## Subjects and methods

### Eligible subjects

The current retrospective, cross-sectional, comparative study was carried out in the Internal Medicine and Pathology Departments at Minia University Hospital, Egypt, in collaboration with the Pathology Department at Minia Oncology Center, Egypt between May 2019 and February 2021. To obtain a power of 99%, a sample size of 40 patients with HCV-related HCC was enrolled in this study. It was calculated at the level of 0.05 significance using G Power 3.19.2 Software. The study was conducted on formalin-fixed paraffin-embedded liver tissues from 40 patients with HCC on a background of HCV-associated liver cirrhosis. This group of patients was compared to two other groups: a group of HCV-related liver cirrhosis patients without HCC, and healthy controls. Data of HCC and cirrhosis patients were retrieved from their medical files in Minia Oncology Center archives. Only patients with adequate liver tissue and complete clinico-pathological data were eligible. The exclusion criteria included causes of chronic liver diseases other than chronic HCV infection, endocrinal diseases that may influence the level of GH-IGF-1 axis, diabetes mellitus, end-organ failure, organ transplantation, hepatic resection, prior locoregional treatment for HCC, extrahepatic and hematological malignancies, autoimmune diseases, as well as steroid and immunosuppressive medications.

### Hepatocellular carcinoma patients (group I)

This group included 40 consecutive patients with HCC, of whom 32(80%) were males. They were recruited from attendants of Minia Oncology Center for liver biopsy. The diagnosis of HCC was based on the characteristic imaging criteria as defined by Bruix and Sherman [45] and histological evaluation [46].

### Liver cirrhosis patients (group II)

This group comprised 40 patients with HCV-related cirrhosis (30(75%) males, and 10(25%) females). They were consecutively enrolled from those referred by outpatient clinics. The diagnosis of liver cirrhosis was built on the standard clinical criteria, in addition to the

histopathological examination [47]. Presence of anti-HCV and detection of serum HCV-RNA for 6 months or more, were characteristic features of chronic HCV infection.

## Healthy controls (group III)

A total of 20 healthy subjects were prospectively collected from subjects who underwent abdominal surgery in the Department of General Surgery at Minia University Hospital. They were 15 (75%) males and 5(25%) females. All were clinically free and showed nothing abnormal in laboratory analyses.

## Informed consent

This study protocol was approved by the Institutional Ethics Committee of the School of Medicine, Minia University, and by Institutional Review Board of Minia Oncology Center, Egypt. The study was performed according to the guidelines and regulations of the 1975 Helsinki Declaration and International Conference on Harmonization Guidelines for Good Clinical Practice. Informed written consent was obtained from all subjects.

## Clinical and laboratory assessment

Demographic, clinical data, and laboratory findings including; the peripheral hemogram, liver and kidney function tests, fasting and postprandial serum glucose levels, plasma levels of alpha fetoprotein (AFP), and virological assays, as well as the imaging studies were obtained by reviewing the medical files. The functional status of the liver was evaluated by the Child-Pugh [48] and the Model for End-Stage Liver Disease (MELD) [49] scoring systems. The clinico-pathological features were assessed according to the Tumor-Node-Metastasis (TNM) [50] and Okuda [51] staging systems. Concerning the healthy volunteers, they were asked to complete a questionnaire on the age, sex, tobacco and alcohol exposure, and current history of any medical illness including diabetes mellitus. They gave venous blood samples to assess the aforementioned laboratory tests using the commercially available kits according to the manufacturer's guidance.

## Immunohistochemistry (IHC)

IHC was performed on 4-μm tissue sections taken from 10% buffered formalin-fixed, paraffin-embedded tissue blocks. Sections were deparaffinized in xylene bath and rehydrated by descending dilutions of ethyl alcohol. Hydrogen peroxide was used to block endogenous peroxidase activity. Antigen retrieval was carried out utilizing citrate buffer concentrate (pH 6). Mouse GHR monoclonal antibody (1/100 dilution, Santa Cruz Biotechnology, Texas, USA), rabbit STAT5 monoclonal antibody (1/100 dilution, Abcam Cambridge Biomedical Campus, UK), mouse IGF-1monoclonal antibody (1/50 dilution, MyBioSource, San Diego USA), mouse Snail-1 monoclonal antibody (1/50 dulation, Santa Cruz Biotechnology Texas, USA), and mouse TGFBR2 monoclonal antibody (1/100 dilution, Santa Cruz Biotechnology, Texas, USA) were used overnight as primary antibodies. Visualization was performed by Avidin-Biotin detection system (DAKO). Positive controls were used to assess correct tissue preparation and staining. One positive control tissue section for each antibody was processed in the same manner as the patient tissue samples and was included in each staining run. One negative control slide was processed for each case by omitting the primary antibody from the staining procedure. Absence of specific staining in the negative control sections was indicative of lack of secondary antibody cross-reactivity with other non-target cellular components.

**Interpretation of GHR, STAT5 and IGF-1 immunoreactivity.** The specimens were evaluated twice in different times by two experienced pathologists, blinded for the clinico-pathological data of the study subjects. The final staining scores of GHR, STAT5, IGF-1, Snail-1, and TGFBR2 were calculated by multiplying the intensity score by the percentage score. The intensity of score was regarded as: absent: 0; weak: 1, moderate: 2, and strong: 3, whereas, the percentage score was graded as follows: none: 0, 1: 1–10%, 2:11–33%, 3:34–66%, and 4: 67–100%. For statistical analysis, a final staining score ≤4 was considered as low expression and a score >4 as a high expression.

## Statistical analyses

Data were analyzed using IBM SPSS for Windows (version 20). Categorical variables were expressed as count and percentages and compared using the Chi-square test and the Fisher exact test when appropriate. One-sample Kolmogorov Smirnov test was used to test for the normality of quantitative variables. The normally distributed variables were described as mean ± standard deviation (SD). Each two groups were compared using Student's t-test, whereas the three groups were compared using one-way analysis of variance followed by Bonferroni post-hoc test between each two groups. The non-normally distributed parameters were expressed as median and interquartile range (IQR) and compared using Kruskal Wallis test followed by Mann Whitney test between each two groups. Pearson's correlation coefficient was used to evaluate the association between two continuous variables, while, Spearman's correlation coefficient was used to test the relation between non-parametric variables. Overall survival (OS) was defined as the time (in months) between the date of disease diagnosis and the date of last follow up or death. Univariate and multivariate survival analysis was done by Cox proportional hazards regression model. Univariate regression models were used to detect independent factors associated with OS. Multivariate Cox analysis was conducted to evaluate meaningful variables detected by univariate analysis. Hazard ratio (HR) and its confidence interval (95% CI) were calculated for each factor. A p value ≤ 0.05 was used as a significant criterion.

## Results

The present study included forty cirrhotic patients with HCC, of whom 32 (80%) were male. The mean age at initial diagnosis was 66 ± 8.1 years. All patients were positive for anti-HCV and PCR for HCV-RNA. Among those patients, 9 (22.5%); 8 (20%), and 23 (57.5%) were class A, B, and C, respectively, according to the Child-Pugh classification. This group of patients was compared to two other groups: a group of HCV-associated cirrhosis patients without HCC and a group of healthy volunteers (c.f., Table 1).

## Cirrhotic patients with HCC versus those without

Group I included 40 cirrhotic patients with HCC, whereas group II consisted of 40 cirrhotic patients without HCC. The baseline characteristics of both groups are depicted in Table 1. There was no statistically significant difference in age, sex, BMI, smoking exposure, platelet count, liver function tests, functional status of the liver, and serum creatinine. However, cirrhotic patients with HCC had significantly higher blood levels of AFP as compared to those without [86(31.8–1000)ng/ml vs. 8(5.1–341.8)ng/ml, p = 0.01]. (c.f., Table 1).

## Cirrhotic patients with HCC versus healthy volunteers

Cirrhotic patients with HCC were matched with healthy volunteers as regards age, sex, smoking exposure, and serum creatinine. The serum levels of total bilirubin, ALT, AST, PT, INR

**Table 1. Baseline characteristics of the study groups.**

| Variable | HCC patients G1 (n = 40) | Cirrhotic patients G2 (n = 40) | Healthy controls G3 (n = 20) | p-value | | | |
|---|---|---|---|---|---|---|---|
| | | | | Among 3 groups | p1 | p2 | p3 |
| Gender [n (%)] | | | | | | | |
| Male | 32 (80%) | 30 (75%) | 15 (75%) | 0.918 | 0.72* | 0.72* | 1* |
| Female | 8 (20%) | 10 (25%) | 5 (25%) | | | | |
| Age (years) [mean ± SD] | 66±8.1 | 62.4±12.1 | 64.3±10.5 | 0.55 | 0.831† | 1† | 1† |
| BMI (kg/m² ) [mean ± SD] | 25.5±3.6 | 24±2.4 | 30±2.6 | **<0.001** | 0.259† | **<0.001†** | **<0.001†** |
| Smoking [n (%)] | | | | | | | |
| No | 28(70%) | 28(70%) | 15(75%) | 0.921 | 1* | 0.72* | 0.72* |
| Yes | 12(30%) | 12(30%) | 5(25%) | | | | |
| Platelets (1×10³/µl) [mean ± SD] | 192±66.4 | 186±49.8 | 248.1±63 | **0.003** | 1† | **0.014†** | **0.006†** |
| Total bilirubin (mg/dl) [median (IQR)] | 1.2(0.8–1.3) | 1.2(0.9–1.3) | 0.2(0.1–0.7) | **0.003** | 0.86†† | **0.01††** | **0.001††** |
| ALT (IU/L) [median(IQR)] | 45.5(41–82) | 43.5(35.5–65.3) | 26(18–30) | **<0.001** | 0.40†† | **<0.001††** | **0.001††** |
| AST(IU/L) [median(IQR)] | 66(39.5–110.8) | 58.5(42.3–82) | 28(25–32) | **<0.001** | 0.39†† | **<0.001††** | **<0.001††** |
| Serum albumin (gm/dl) [mean ± SD] | 3.4±0.7 | 3.7±0.7 | 3.9±0.3 | **0.04** | 0.30† | **0.05†** | 0.62† |
| PT (seconds) [mean ± SD] | 14.8±2.8 | 13.6±2.7 | 11.7±1.1 | **0.041** | 1† | **0.04†** | 0.264† |
| INR [mean ± SD] | 1.3±0.3 | 1.2±0.3 | 1.1±0.1 | **0.036** | 1† | **0.03†** | 0.320† |
| Serum creatinine (mg/dl) [mean ± SD] | 0.8±0.3 | 0.8±0.2 | 0.7±0.1 | 0.150 | 1† | 0.831† | 0.159† |
| AFP (ng/dl) [median (IQR)] | 86(31.8–1000) | 8(5.1–341.8) | 4(3–4.8) | **<0.001** | **0.01††** | **<0.001††** | **<0.001††** |
| Child-Pugh class [n (%)] | | | | | | | |
| A | 9(22.5%) | 5(12.5%) | --- | | 0.071* | ----- | ----- |
| B | 8(20%) | 10(25%) | --- | | | | |
| C | 23(57.5%) | 25(62.5%) | ---- | | | | |
| Child-Pugh score [mean ± SD] | 9.8±2.3 | 9.8±2.3 | ---- | | 0.60§ | ----- | ----- |
| MELD [median (IQR)] | 9.5(7–12.5) | 8(7–10.8) | ---- | | 0.35†† | ----- | ----- |
| TNM score (I+II/III+IV)[n.(%)] | 24(60%)/16(40%) | ---- | ----- | | ----- | ----- | ----- |
| Okuda score (1/2+3) [n.(%)] | 22(55%)/18(45%) | ---- | ----- | | ----- | ----- | ----- |

HCC: hepatocellular carcinoma; G: group; n: number; BMI: body mass index; kg/m²: kilogram/meter²; ALT: alanine aminotransferase; AST: aspartate transaminase; PT: prothrombin time; INR: international normalized ratio; AFP: alpha-fetoprotein; MELD: model of end stage liver disease; TNM: Tumor-Node-Metastasis

*: Chi square test

†: one way analysis of variance test followed by Bonferroni post-hoc test between each two groups

††: Kruskal Wallis test followed by Mann Whitney test between each two groups

§: Student's t-test; SD: standard deviation; IQR: interquartile range; p1: G1 vs. G2; p2: G1 vs. G3; p3: G2 vs. G3.

Bold values denote significant results

and AFP were significantly higher in HCC patients than in healthy controls [1.2(0.8–1.3)mg/dl vs.0.2(0.1–0.7) mg/dl, p = 0.01; 45.5(41–82)IU/L vs. 26(18–30)IU/L, p<0.001; 66(39.5–110.8) IU/L vs. 28(25–32)IU/L, p<0.001; 14.8±2.8sec. vs. 11.7±1.1sec., p = 0.04; 1.3±0.3 vs. 1.1±0.1, p = 0.03, and 86(31.8–1000)ng/ml vs.4(3–4.8)ng/ml, p<0.001, respectively. Whilst, they displayed statistically significant lower values of BMI, platelet count, and serum albumin versus healthy control group (25.5 ± 3.6 kg/m² vs. 30 ±2.6 kg/m², p<0.001; 192±66.4 (1×10³ µl) vs. 248.1±63 (1×10³ µl), p = 0.014, and 3.4±0.7 gm/dl vs. 3.9±0.3 m/dl, p = 0.05, respectively (c.f., Table 1).

## Comparison of hepatic expression of GHR, STAT5, IGF-1, Snail-1 and TGFBR2 proteins in the study groups

In cirrhotic patients with HCC, hepatic expression of GHR was significantly lower than that in cirrhotic patients without HCC (0.5(0–3.8) vs. 4(0.8–6), p = 0.02), and control group (0.5(0–3.8) vs. 6 (4–7), p<0.001). Similar trends were observed in STAT5 and IGF-1 hepatic expression. Both were significantly lower in cirrhotic patients with HCC in comparison to those without (0.9(0–2) vs. 1.5 (0–4), p = 0.02) for STAT5, and (1.5(0–6) vs. 4(3–6), p = 0.048 for IGF-1, and healthy controls (0.9 (0–2) vs. 4(2–5.5), p<0.001), and 1.5(0–6) vs. 6(4.5–9), p<0.001), respectively (c.f., Table 2).

The hepatic expression of Snail-1 was found to be significantly higher in cirrhotic patients with HCC compared to those without HCC (4(3–6) vs. 2(1–3), p<0.001) and control group (4 (3–6) vs. 1(1–2), p<0.001), whereas cirrhotic patients with HCC had significantly lower hepatic expression of TGFBR2 than did cirrhotic patients without HCC (2.5(2–4) vs. 4(3–6), p = 0.046) and healthy controls (2.5(2–4) vs. 6(3–9), p = 0.019) (c.f., Table 2).

Regarding the subcellular localization of GHR and IGF-1 expression, it was predominantly cytoplasmic in HCC patients. Whereas, STAT5 was found in the cytoplasm in 22(55%) cases, localized to the nucleus in 13(32.5%) cases and to both sites in 5(12.5%) cases (c.f., Fig 1A–1I).

Snail-1 hepatic expression was mainly cytoplasmic in HCC patients, although it was mostly nuclear in case of TGFBR2 expression (c.f., Fig 2A–2F).

Moreover, we found that the expression of all these proteins in stromal cells of HCC microenvironment; including: kupffer cells (KCs), hepatic stellate cells (HSCs), liver sinusoidal endothelial cells (LSECs), Cholangiocytes, and stromal inflammatory cells was higher in HCC patients when compared to cirrhotic patients. The differences yielded statistical significance when LSECs were examined for expression of GHR (70% vs. 25%, p<0.01), STAT5 (80% vs. 15%, p<0.001) and TGFBR2 (95% vs. 10%, p>0.001). In the case of HSCs, expression of Snail-1 and TGFBR2 was significantly higher in HCC patients than cirrhotic patients (60% vs. 20%, p = 0.022, and 75% vs. 15%, p< 0.001, respectively). Similar trend was observed in stromal inflammatory cells (90% vs. 25%, p< 0.001 for Snail-1 and 90% vs. 4%, p = 0.002 for TGFBR2) (data not shown) (c.f., Fig 3A–3N).

## Correlations of hepatic expression of GHR/STAT5/IGF-1 signaling pathway with different clinico-biochemical parameters, Snail-1, and TGFBR2 in cirrhotic patients

In cirrhotic patients, negative correlations were found between hepatic expression of GHR on the one hand and patient age (r = - 0.30; p = 0.03) and AFP (r = 0.31; p = 0.029) on the other

**Table 2. Comparison of hepatic expression of GHR, STAT5, IGF-1, Snail-1 and TGFBR2 proteins in the study groups.**

| Variable | HCC patients G1(n = 40) | Cirrhotic patients G2(n = 40) | Healthy controls G3(n = 20) | p-value | | | |
|---|---|---|---|---|---|---|---|
| | | | | Among 3 groups | p1 | P2 | P3 |
| Hepatic expression: | | | | | | | |
| GHR [median (IQR)] | 0.5(0–3.8) | 4(0.8–6) | 6(4–7) | <0.001 | **0.02** | **<0.001** | **0.01** |
| STAT5 [(median (IQR)] | 0.9(0–2) | 1.5(0–4) | 4(2–5.5) | 0.003 | **0.02** | **<0.001** | **0.03** |
| IGF-1[(median (IQR)] | 1.5(0–6) | 4(3–6) | 6(4.5–9) | 0.001 | **0.048** | **<0.001** | **0.025** |
| Snail-1 [(median (IQR)] | 4(3–6) | 2(1–3) | 1(1–2) | <0.001 | **<0.001** | **<0.001** | 0.228 |
| TGFBR2 [(median (IQR)] | 2.5(2–4) | 4(3–6) | 6(3–9) | 0.028 | **0.046** | **0.019** | 0.213 |

HCC: hepatocellular carcinoma; G: group; n: number; GHR: growth hormone receptor; STAT5: signal transducer and activator of transcription 5; IGF-1: insulin like growth factor-1, TGFBR2: type2 transforming growth factor-beta receptor. Data are expressed as median (interquartile range) and compared using Kruskal Wallis test among the three groups followed by Mann Whitney test between each two groups; p1: G1 vs. G2; p2: G1 vs. G3; p3: G2 vs. G3.
Bold values denote significant results

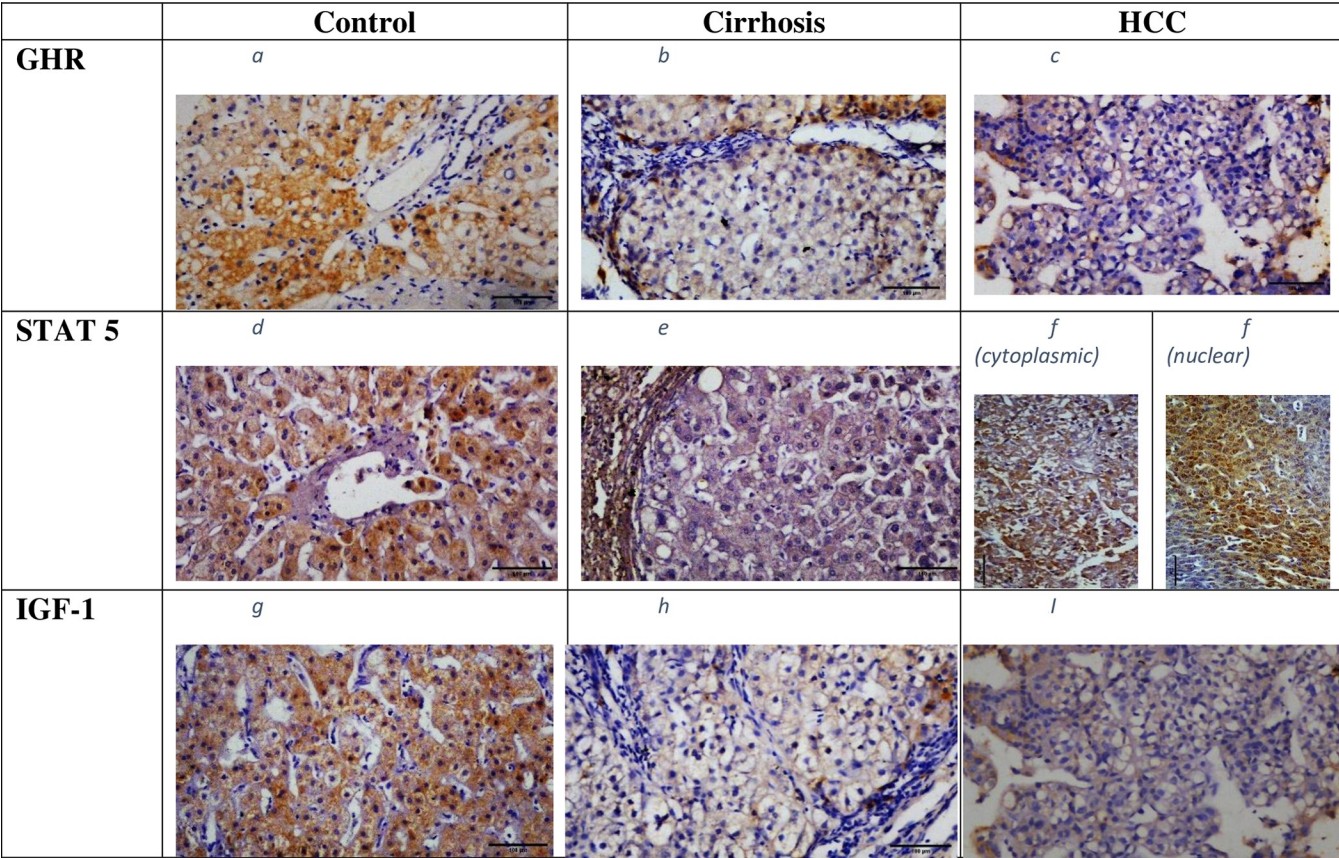

**Fig 1.** (a-i): Hepatic expression of GHR, STAT5, and IGF-1 proteins in the study groups. GHR expression: a)in healthy control: b) in cirrhotic patients; c) in HCC patients; (predominantly cytoplasmic); STAT5 expression: d) in healthy control; e) in cirrhotic patients; f) in HCC patients (cytoplasmic/nuclear) IGF-1 expression: g)in healthy controls; h) in cirrhotic patients; i) in HCC patients (predominantly cytoplasmic). Magnification 200 X scale bar 100 μm. HCC: hepatocellular carcinoma; GHR: growth hormone receptor; STAT5: signal transducer and activator of transcription5; IGF-1: insulin like growth factor-1.

one. Also a significant negative correlation was detected between hepatic expression of STAT5 and blood levels of AFP (r = - 0.29; p = 0.04). Regarding the hepatic expression of IGF-1, it was correlated positively with patient BMI (r = 0.33; p = 0.011) and negatively with patient age (r = - 0.31; p = 0.035), serum levels of liver enzymes (r = - 0.28; p = 0.03 for ALT, and r = - 0.34; p = 0.008 for AST), PT (r = - 0.27; p = 0.036), as well as serum creatinine (r = - 0.33; p = 0.013). It was notable that the hepatic expression of the three studied proteins showed significant positive correlations between them. As regards the inducers of EMT, hepatic expression of Snail-1was correlated negatively with expression of GHR, STAT5, and IGF-1 proteins (r = -0.55, p = 0.02; r = -0.47, p = 0.035, and r = -0.51, p = 0.009, respectively). On the contrary, there were positive correlations between hepatic expression of TGFBR2 and expression of all these proteins (r = 0.47, p = 0.034; r = 0.49, p = 0.023, and r = 0.57, p<0.001, respectively) (c.f., Table 3).

## Relationship between hepatic expression of GHR/STAT5/IGF-1 signaling pathway and clinico-pathological features of the tumor in HCC patients

HCC patients with lower GHR hepatic expression exhibited significantly higher frequency of patients with AFP> 100 ng/ml (76.5% vs. 23.5%; p = 0.048); tumor size >5cm (82% vs. 18%; p = 0.02); vascular invasion (85.5% vs. 14.5%; p = 0.002), and advanced TNM stage (80.6% vs.

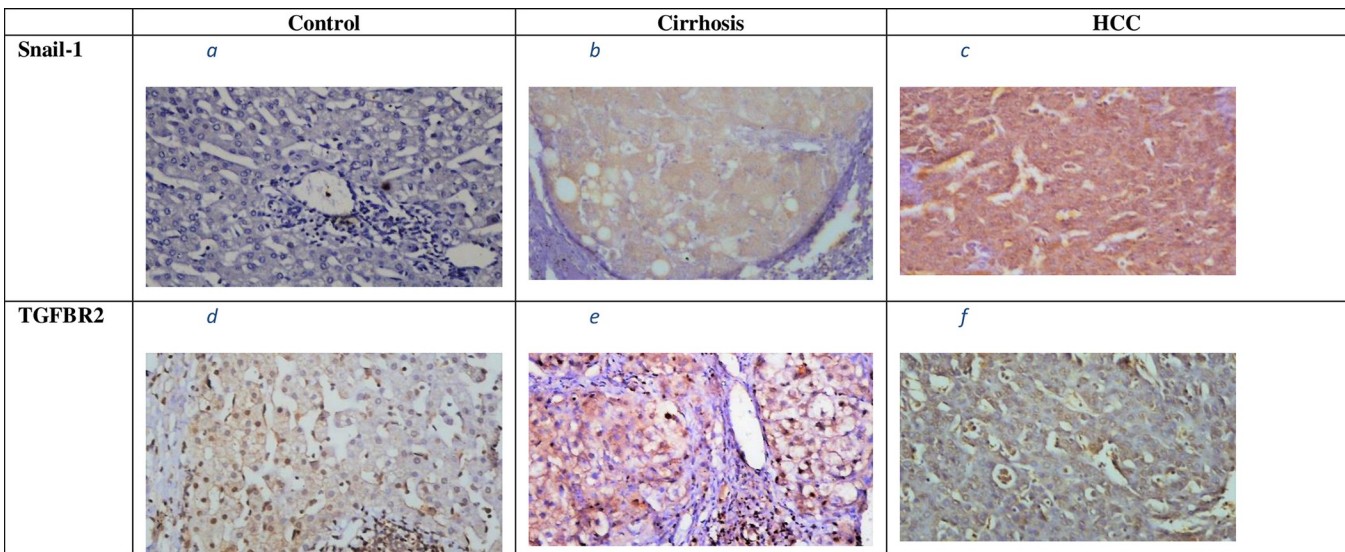

**Fig 2.** (a-f): Hepatic expression of Snail-1 and TGFBR2 proteins in the study groups. Snail-1expression: a) in healthy control: b) in cirrhotic patients; c) in HCC patients; (predominantly cytoplasmic); TGFBR2 expression: d) in healthy control; e) in cirrhotic patients; f) in HCC patients (predominantly nuclear). Magnification 200 X scale bar 100 μm. HCC: hepatocellular carcinoma; TGFBR2: type 2 transforming growth factor-beta receptor.

19.4%; p = 0.01). Also those with lower hepatic expression of STAT5 had significantly higher proportion of patients with AFP >100 ng/ml (84.8% vs. 15.2%; p = 0.006); vascular invasion (86% vs. 14; p = 0.009, and advanced pathological stage (86.7% vs. 13.3%; p = 0.007). A significantly higher proportion of patients with portal vein infiltration was found among HCC patients with lower IGF-1 hepatic expression (93.75% vs. 6.25%; p<0.001) (c.f., Table 4).

## Factors associated with overall survival in HCC patients

Herein, univariate analysis by Cox hazard model showed that age>60 years (HR (95% CI) = 2.34 (1.15–3.86); p = 0.07), vascular invasion (HR (95%CI) = 2.19(1.19–2.82); p = 0.0001); advanced TNM stage (HR (95% CI) = 3.10 (1.61–5.12); p = 0.0001), lower hepatic expression of GHR (HR (95%CI) = 3.8(1.98–5.67); p<0.0001); lower hepatic expression of STAT5 (HR (95% CI) = 1.71(1.22–2.43); p<0.01, and lower hepatic expression of IGF-1 (HR(95%CI) = 2.3 (1.27–2.99); p = 0.0061) were significantly related to worse OS in HCC patients. Age > 60 years (HR(95%CI) = 2.0(1.03–3.3); p = 0.037, vascular invasion (HR(95%CI) = 3.15(1.89–10.61); p = 0.001, advanced TNM stage (HR(95%CI) = 5.32(1.63–12.43); p = 0.0014; lower GHR expression (HR(95% CI) = 3.1 (1.43–4.76); p = 0.0004), lower hepatic expression of STAT5 (HR(95%CI) = 1.45(0.96–2.33); p = 0.046) and lower hepatic expression of IGF-1 (HR (95%CI) = 1.26(0.75–2.11); p = 0.049) were found to be independent predictors of mortality in those patients when multivariate Cox analysis was used (c.f., Table 5).

## Discussion

In spite of the progress in the management of HCC, the clinical outcome remains inadequate due to the complexity and heterogeneity of the disease, the rarity of treatment options, being diagnosed at an advanced stage, as well as, the increased rate of recurrence and distant metastasis [52]. Therefore, the recognition of the exact mechanisms which may play a role in the tumor incidence and progression, could be helpful to represent new targeted treatment approaches of HCC.

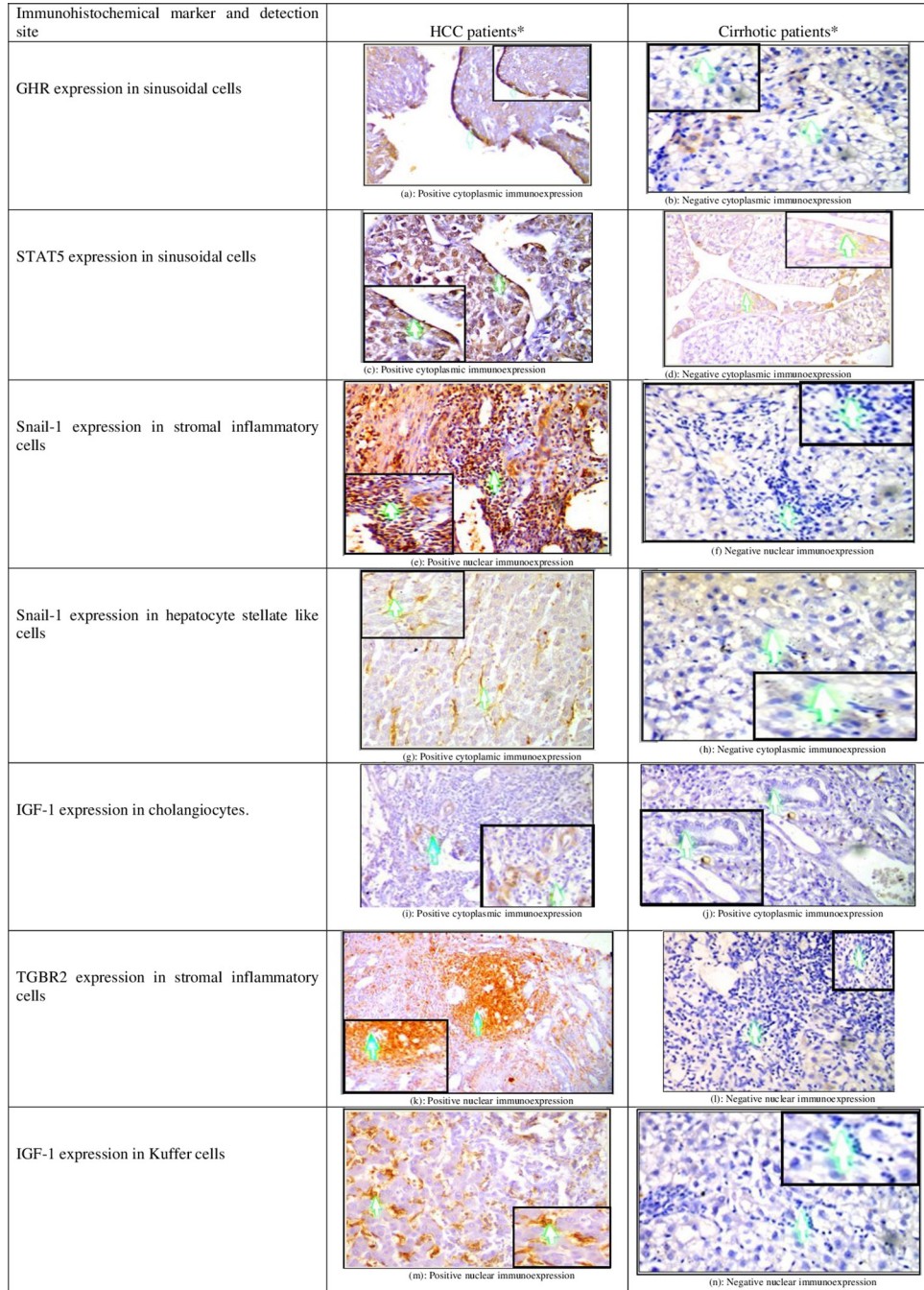

**Fig 3.** (a-n): Examples of expression of GHR, STAT5, IGF-1, Snail-1 and TGFBR2 in a variety of stromal cells of TME in HCC and cirrhotic patients. GHR: growth hormone receptor, STAT5: signal transducer and activator of transcription5; IGF-1: insulin like growth factor-1; TGFBR2: type 2 transforming growth factor-beta receptor; TME: tumor microenvironment; HCC: hepatocellular carcinoma. *: Images are presented at 20X magnification power with 200X zoom in boxes.

Recently, there has been increasing evidences which explore the role of GH-STAT5-1GF-1 axis in cancer development [53]. Binding of GH to its cognate receptor, is an essential step to exert its effects. In the present study, we observed that HCC patients exhibited significantly lower hepatic expression of GHR than those of cirrhosis and control groups, mostly due to

**Table 3. Correlations of hepatic expression of GHR/STAT5/IGF-1 signaling pathway with different clinico-biochemical parameters, Snail-1 and TGFBR2 in cirrhotic patients.**

| Variable | GHR protein expression | | STAT5 protein expression | | IGF-1 protein expression | |
|---|---|---|---|---|---|---|
| | R | p-value | r | p-value | r | p-value |
| STAT5 hepatic expression | 0.35 | **0.001**† | ------- | -------- | ------- | ------- |
| IGF-1 hepatic expression | 0.45 | **0.02**† | 0.53 | **0.0001**† | ----- | ----- |
| Age (years) | - 0.30 | **0.03**† | -0.15 | 0.24† | -0.31 | **0.035**† |
| BMI (kg/m$^2$) | 0.20 | 0.12† | 0.11 | 0.41† | 0.33 | **0.011**† |
| Smoking (Yes/No) | -0.09 | 0.47†† | -0.05 | 0.72†† | -0.08 | 0.53†† |
| Platelets (1×10$^3$/μl) | 0.01 | 0.96† | -0.10 | 0.44† | 0.09 | 0.50† |
| Total bilirubin (mg/dL) | -0.07 | 0.61† | -0.09 | 0.47† | 0.07 | 0.59† |
| ALT (IU/L) | -0.05 | 0.78† | -0.05 | 0.72† | -0.28 | **0.03**† |
| AST (IU/L) | -0.22 | 0.36† | -0.09 | 0.49† | -0.34 | **0.008**† |
| Serum albumin (gm/dl) | 0.17 | 0.20† | 0.14 | 0.29† | 0.14 | 0.30† |
| Prothrombin time (seconds) | -0.22 | 0.09† | -0.07 | 0.59† | -0.27 | **0.036**† |
| INR | -0.12 | 0.38† | -0.14 | 0.28† | -0.18 | 0.231† |
| Serum creatinine (mg/dl) | -0.06 | 0.63† | -0.02 | 0.89† | -0.38 | **0.013**† |
| Alpha-fetoprotein (ng/ml) | -0.31 | **0.029**† | -0.29 | **0.04**† | -0.23 | 0.08† |
| Child-Pugh score | -0.06 | 0.70† | -0.21 | 0.06† | -0.02 | 0.92† |
| MELD score | -0.05 | 0.75† | -0.15 | 0.36† | -0.03 | 0.88† |
| Snail-1 | -0.55 | **0.02**† | -0.47 | **0.035**† | -0.51 | **0.009**† |
| TGFBR2 protein | 0.47 | **0.034**† | 0.49 | **0.023**† | 0.57 | **<0.001**† |

GHR: growth hormone receptor; STAT5: signal transducer and activator of transcription 5; IGF-1: insulin like growth factor-1; r: correlation coefficient; BMI: body mass index; kg/m$^2$: kilogram/meter$^2$; ALT: alanine aminotransferase; AST: aspartate transaminase; INR: international normalized ratio; MELD: model of end stage liver disease; TGFBR2: type2 transforming growth factor-beta receptor

†: Pearson's correlation

††: Spearman's correlation

Bold values denote significant results

hepatocellular dysfunction as a consequence of chronic liver disease, and tumor burden [54]. Our findings were compatible with the scarcely available studies in literature [55–57], albeit, GHR hepatic expression in these studies were performed by different techniques. It has been revealed that a lower GHR levels have led to a state of GH resistance which may be related to decreased GH clearance, defective synthesis of IGF-1 as a result of hepatocellular damage, and defective binding capability of GH to GHR [58]. GH mediates its pro-oncogenic effects by creating a protumorigenic milieu via various mechanisms. Elevated GH leads to accumulation of unrepaired damaged DNA, which results in increasing chromosomal instability and oncogenic aberrations [59]. GH may increase the propensity of cancer development by suppressing many tumor suppressor genes such as p53, leading to promotion of cell proliferation, and augmentation of GH role on accumulation of damaged DNA [60]. Moreover, increased GH may trigger the process of EMT by suppressing the expression of E-cadherin, which is a cell-to-cell adhesion molecule [61]. Additionally GH assists cell motility and invasion, as well as, acquiring cancer stem cell-like criteria of HCC cells, by inhibiting another tight junction protein called Claudin-1 through activation of STAT3 in HCC [62]. The effect of GH on EMT has been reported in HCC patients [63]. A significant overexpression of Snail-1 which represents a critical transcriptional repressor of E-cadherin [34], was noticed in HCC patients versus both control groups in the current study. Loss of functionality of E-cadherin has been found in different malignancies including the liver [64]. Although the distinct role of Snail-1 in triggering hepatocarcinogenesis is not wholly clear, a lot of evidence points to its vital role during

**Table 4. Relationship between hepatic expression of GHR/STAT5/IGF-1 signaling pathway and clinico-pathological features of the tumor in cirrhotic patients with hepatocellular carcinoma.**

| Variable | Hepatic expression of GHR | | | | Hepatic expression of STAT5 | | | | Hepatic expression of IGF-1 | | | |
|---|---|---|---|---|---|---|---|---|---|---|---|---|
| | n. | No/low n(%) | High n(%) | p-value | n. | No/low n(%) | High n(%) | p-value | n. | No/low n(%) | High n(%) | p-value |
| Age (years) | | | | | | | | | | | | |
| ≤ 60 | 20 | 14(70) | 6(30) | 1 | 20 | 8(40) | 12(60) | 1 | 20 | 12(60) | 8(40) | 1 |
| >60 | 20 | 12(60) | 8(40) | | 20 | 6(30) | 14(60) | | 20 | 14(70) | 6(30) | |
| Gender | | | | | | | | | | | | |
| Male | 22 | 14(63.6) | 8(36.4) | 1 | 36 | 34(94.4) | 2(5.6) | 1 | 34 | 22(64.7) | 12(35.3) | 1 |
| Female | 18 | 12(66.7) | 6(33.3) | | 4 | 4(100) | 0(0) | | 6 | 4(66.7) | 2(33.3) | |
| Child-Pugh score | | | | | | | | | | | | |
| A | 28 | 20(71.4) | 8(28.6) | 0.613 | 28 | 10(35.7) | 18(64.3) | 1 | 28 | 14(50) | 14(50) | 1 |
| B+C | 12 | 6(50) | 6(50) | | 12 | 4(33.3) | 8)66.7) | | 12 | 6(50) | 6(50) | |
| MELD score | | | | | | | | | | | | |
| ≤12 | 20 | 16(80) | 4(20) | 0.350 | 20 | 8(40) | 12(60) | 1 | 20 | 10(50) | 10(50) | 1 |
| >12 | 20 | 10(50) | 10(50) | | 20 | 6(30) | 14(70) | | 20 | 10(50) | 10(50) | |
| AFP (ng/ml) | | | | | | | | | | | | |
| ≤100 | 6 | 2(33.3) | 4(66.7) | **0.048** | 7 | 2(28.6) | 5(71.4) | **0.006** | 20 | 10(50) | 10(50) | 1 |
| >100 | 34 | 26(76.5) | 8(23.5) | | 33 | 28(84.8) | 5(15.2) | | 20 | 10(50) | 10(50) | |
| Tumor number | | | | | | | | | | | | |
| Single | 24 | 18(75) | 6(25) | 0.356 | 24 | 8(33.3) | 16(66.7) | 1 | 24 | 14(58.3) | 10(41.7) | 0.650 |
| Multiple | 16 | 8(50) | 8(50) | | 16 | 6(37.5) | 10(62.5) | | 16 | 6(37.5) | 10(62.5) | |
| Tumor size (cm) | | | | | | | | | | | | |
| ≤5 | 12 | 5(41.7) | 7(58.3) | **0.02** | 20 | 8(40) | 12(60) | 1 | 20 | 10(50) | 10(50) | 1 |
| >5 | 28 | 23(82) | 5(18) | | 20 | 6(30) | 14(70) | | 20 | 10(50) | 10(50) | |
| Vascular invasion | | | | | | | | | | | | |
| No | 12 | 4(33.3) | 8(66.7) | **0.002** | 11 | 5(45.5) | 6(54.5) | **0.009** | 8 | 2(25) | 6(75) | **<0.001** |
| Yes | 28 | 24(85.5) | 4(14.5) | | 29 | 25(86) | 4(14) | | 32 | 30(93.75) | 2(6.25) | |
| Lymphatic permeation | | | | | | | | | | | | |
| No | 36 | 24(66.7) | 12(33.3) | 1 | 36 | 12(33.3) | 24(66.7) | 1 | 36 | 18(50) | 18(50) | 1 |
| Yes | 4 | 2(50) | 2(50) | | 4 | 2(50) | 2(50) | | 4 | 2(50) | 2(50) | |
| T.N.M stage | | | | | | | | | | | | |
| I | 9 | 3(33.3) | 6(66.7) | **0.01** | 10 | 4(40) | 6(60) | **0.007** | 24 | 14(58.3) | 10(41.7) | 0.650 |
| II-IV | 31 | 25(80.6) | 6(19.4) | | 30 | 26(86.7) | 4(13.3) | | 16 | 6(37.5) | 10(62.5) | |
| Okuda stage | | | | | | | | | | | | |
| 1 | 22 | 14(63.6) | 8(36.4) | 1 | 22 | 10(45.5) | 12(54.5) | 0.374 | 22 | 12(54.5) | 10(45.5) | 1 |
| 2+3 | 18 | 12(66.7) | 6(33.3) | | 18 | 4(22.2) | 14(77.8) | | 18 | 8(44.4) | 10(55.6) | |

Number of patients = 40

n.: number; GHR: growth hormone receptor; STAT5: signal transducer and activator of transcription 5; IGF-1: insulin like growth factor-1; MELD: model of end stage liver disease; AFP: alpha-fetoprotein; TNM: Tumor-Node-Metastasis. Data are expressed as proportions and percentages, and compared using Chi-square statistic or Fisher's exact test

Bold values denote significant results

malignant transformation. The tumor suppressor gene p53 hinders tumor cell invasion through the degeneration of Snail protein in HCC [65]. Co-operatively, Notch 1 and reactive oxygen species increase the level of Snail protein in hepatoma cells via induction of phosphori-nositide 3- Kinase/Akt signaling pathway [66]. Yuan et al., [67] found that long non-coding UCID interacts with Snail protein to increase its stability which promts the EMT process to hasten HCC metastasis. Contrariwise, downregulated hepatic expression of TGFBR2 was seen

**Table 5. Univariate and multivariate analyses of risk factors related to overall survival in cirrhotic patients with hepatocellular carcinoma.**

| Variable | Overall survival | | | |
|---|---|---|---|---|
| | Univariate | | Multivariate | |
| | HR (95%CI) | p-value | HR (95%CI) | p-value |
| Age (years) (≤ 60vs>60) | **2.34(1.15–3.86)** | **0.007** | **2.0(1.03–3.3)** | **0.037** |
| Gender (Male vs. Female) | 0.94(0.8–1.13) | 0.541 | | |
| Child-Pugh score (A vs. B+C) | 1.11(0.54–1.83) | 0.675 | | |
| MELD score (>12 vs. ≤12) | 1.32(0.69–2.14) | 0.285 | | |
| AFP (ng/ml) (≤100vs.>100) | 1.59(0.81–3.12) | 0.273 | | |
| Tumor number (single vs. multiple) | 0.58(0.42–1.19) | 0.269 | | |
| Tumor size (cm) (≤5 vs.>5) | 1.37(0.65–2.43) | 0.269 | | |
| Vascular invasion (No vs. Yes) | **2.19(1.19–2.82)** | **0.0001** | **3.15(1.89–10.61)** | **0.001** |
| Lymphatic permeation (No vs. Yes) | 1.12(0.63–1.87) | 0.663 | | |
| TNM stage (I vs. II-IV) | **3.10(1.11–5.12)** | **0.0001** | **5.32(1.63–12.43)** | **0.0014** |
| Okuda stage (1vs. 2+3) | 0.83(0.49–1.51) | 0.539 | | |
| GHR expression (0–4 vs. 5–12) | **3.8(1.98–5.67)** | **<0.0001** | **3.1(1.43–4.76)** | **0.0004** |
| STAT5 expression (0–4 vs. 5–12) | **1.71(1.22–2.43)** | **<0.01** | **1.45(0.96–2.33)** | **0.047** |
| IGF-1 expression (0–4 vs. 5–12) | **2.3(1.27–2.99)** | **0.0061** | **1.26(0.75–2.11)** | **0.049** |

Number of patients = 40

HR: hazard ratio; CI: confidence interval; MELD: model of end stage liver disease; AFP: alpha-fetoprotein; TNM: Tumor-Node-Metastasis; GHR: growth hormone receptor; STAT5: signal transducer and activator of transcription 5; IGF-1: insulin like growth factor-1.

Bold values denote significant results

in our HCC patients compared with the two control groups. Reduced TGFBR2 expression might result in failed cell growth arrest regulated by TGF-β which accelerates the cell oncogenesis [40]. Down-regulation of TGFBR2 is mostly due to epigenetic silencing by promotor methylation [68], in addition to TGFBR2 gene loss or mutations which are rare. Our results were consistent with those found by other investigators [44], however, others showed unchanged TGFBR2 expression in HCC patients [69]. Interestingly, we found that GHR expression negatively correlated to Snail-1 expression, but was positively correlated to TGFBR2, with statistical significance.

In HCV-related HCC patients, HCV infection leads to a production of various inflammatory and fibrotic mediators such as proinflammatory cytokines, cell death signals, and reactive oxygen species [70], as well as, hepatic stellate cells activation [71]. All these events, creates a cirrhotic microenvironment that refers to the "field cancerization", which initiates and promotes hepatocarcinogenesis [72] and probably dysregulates GHR expression. The viral core protein down-regulates CDKN1A; one of the cell cycle inhibitors leading to an aggressive type of HCC [73]. By contrast, a previous study revealed an increased expression of GHR in HCC patients as compared to healthy controls [74]. It has been reported that binding of GH to the atypically expressed GHR activates the JAK2 pathway, induces EMT, and promotes tumorigenesis [75].

Herein, we reported a significant positive correlation between hepatic expression of both GHR and IGF-1 proteins, whereas there was significant negative correlations between hepatic expression of GHR on the one hand and patient age, serum levels of AFP, vascular invasion and the TNM stage of hepatoma on the other. Also decreased hepatic expression of GHR was associated with increased serum levels of AFP, large tumor size, vascular invasion, advanced histo-pathological stage, and worse outcome. Nearly similar relations were reported by Lin et al. [57].

Many studies have been mentioned the critical roles of STAT5 proteins in the development of various solid tumors such as prostate [76], colorectal [77], breast [78], and lung [79] cancers, as well as, hematological malignancies [80]. However, studies on the role of STAT5 in HCC proved that STAT5 poses contrasting functions in different contexts, where it can act as a tumor suppressor [81, 82], or as an oncogene in other situation [83–85]. For the first time, the current study showed lower hepatic expression of STAT5 in HCC patients than cirrhotic patients and healthy volunteers, as clinical studies in this regard remains lacking. On the contrary, other investigators [83–85] reported enhanced expression of STAT5 in HCC cells that was frequently associated with tumor aggressiveness and poor clinical outcome. The oncogenic role of STAT5 in these studies was mediated through induction of cell growth and proliferation, recruitment of cancer stem cells, promotion of drug.-resistance, as well as epithelial meseuchymal transition. Furthermore, Lee et al. [83] mentioned the role of hepatitis B virus X protein in activating STAT5b in HCC patients. Meanwhile, loss of STAT5 is associated with hepatocarcinogenesis as a result of increased oncogenic STAT3 activity and induction of liver fibrosis via increased levels of TGF-β [86]. Moreover, STAT5 deficiency leads to: increased activity of oncogenic JNK1 and c-Jun (stress-activated protein kinases) [87]; attenuated expression of tumor suppressor gene p53 [88], and decreased activity of glutathione S-transferases which stimulates oxidative stress and hepatocellular damage [89]. As regard the oncogenic effects of STAT3, it is well established that activation of STAT3 promotes cell proliferation [90], and suppresses apoptosis via up-regulation of anti-apoptotic proteins [91], and down-regulation of pro-apoptotic genes [92]. It induces angiogenesis by enhancing the expression of various pro-angiogenic molecules such as vascular endothelial growth factor in the tumor microenvironment [93]. Moreover, activated STAT3 motivates the secretion of many chemokines and cytokines such as, IL-6 and IL-16 to maintain activation of immune cells in the tumor tissue [94]. STAT3 activation also regulates the expression of different cancer cells; namely CD 44 [95], and CD 133 [96] positive cells which maintain the stem cell-like characteristics in HCC cells by inducing the Notch signaling pathway [97]. Furthermore STAT3 is responsible for providing the tumor cells with the energetic demands [98]. Also it assists the motility and invasion of HCC cells by adjusting the expression of matrix metalloproteinases that cleave the extracellular matrix in the tumor microenvironment [99].

Although the activated STAT proteins localize in the nucleus [15], in the current study, STAT5 was predominantly cytoplasmic. It has been reported that STAT5 is found in both the nucleus and the cytoplasm of the cells depending upon dynamic trafficking. Its nuclear entry is induced by the importin-α3/β1 system, that is completely independent of its activation status. It continuously shuttles in and out of the nucleous through chromosome region maintenance 1-dependent and- independent pathways [100], thus unphosphorylated STAT5 could be elicited in the nucleus. Moreover, activated STAT5 may be exclusively located in the cytoplasm by inducing Scr family Kinases that leads to cytoplasmic retention of activated STAT 5 via interaction of the SH2 domain [101]. So, the oncogenic STAT5 activity may include cytoplasmic function in addition to the nuclear function through crosstalk with various signaling pathways.

The hepatic expression of STAT5 in our HCC patients was found to be inversely correlated with the tumor size, vascular invasion, TNM tumor stage, and serum level of AFP. Furthermore, lower expression of STAT5 was significantly associated with serum AFP > 100 ng/ml, vascular invasion, advanced tumor stage, and poor patient prognosis. Regarding the relation between low hepatic expression of STAT5 and vascular invasion in our HCC patients, development of HCC coexists with persistent inflammatory cells; including tumor-associated macrophages (TAM) [102]. TAMs promote EMT by producing factors such as IL-6, IL-8, TGF-β, as well as matrix metalloproteinase 2 and 9 which break down extracellular matrix to assist local

invasion and metastasis of tumor cells [103]. IL-6 binds to its receptor and interacts with JAK2 leading to STAT3 activation which triggers EMT that is transcriptionally induced by Twist [104]. IL-8 is another cytokine that is secreted by TAMs in HCC; it has a role in tumor growth, survival, angiogenesis, as well as EMT via the JAK2/STAT3/Snail signaling pathway [105] and activation of CXC chemokine receptor, and CC chemokine ligand 2 [106]. Overexpression of TGF-β that secreted by TAMs interacted with different transcription factors such as Snail and Slug to induce EMT in tumor cells through down-regulation of E-cadherin expression, and up-regulation of vimentin expression [107]. Although it has been proved by RNA-sequencing analysis that STAT5 genes related to the anti-tumor immune response in TAM, Jesser et al. [108] found that loss of STAT5 in macrophages increased its ability to express tumor-promoting factors which enhanced the tumor cell migration and metastasis in vitro and in vivo. Importantly, our results showed that STAT5 expression in HCC patients was negatively correlated with Snail-1 expression, but directed to a positive correlation with TGFBR2 expression.

Another interesting finding of this study; we revealed that tissue expression of IGF-1 protein was significantly lower in HCC patients than cirrhotic patients and healthy controls. Our findings were consistent with those of previous studies [57, 109–111], but only as compared to healthy volunteers. Reduced hepatic expression of IGF-1 in HCC is multifactorial. Reduced levels of IGF-1 lead to development of GH resistance [56], decreased expression of GHR [112] due to hepatocellular damage by tumor cells [113], as well as, existence of feedback circuit at endocrine [114] and paracrine [115] loops. In this study, the tumor in HCC patients was arising on a background of HCV-associated liver cirrhosis. A lot of studies have demonstrated that liver cirrhosis by itself is responsible for decreased serum concentrations [116–118] and tissue expression [112, 119, 120] of IGF-1 protein, that have been more pronounced in patients with HCV infection than those without [20]. IGF-1 deficiency in cirrhotic patients may be related to hepatocellular dysfunction, malnutrition, oxidative damage, altered lipid metabolism, and insulin resistance [121, 122]. HCV infection triggers the expression of glucose 6-phosphatase and phosphoenol-pyruvate carboxykinase 2 leading to enhanced glucose production. Additionally, HCV infection negatively regulates the expression of glucose transporter-type 4 that is responsible for glucose uptake; therapy producing a state of insulin resistance [123], which leads to acceleration of liver fibrosis [124], and hepatocellular dysfunction that is usually followed by IGF-1 deficiency. IGF-1 deficiency has various detrimental effects. In physiological situation, liver is not a target organ for IGF-1 due to absence of its receptor in hepatocytes [125]. IGF-1 deficiency results in hepatic expression of genes encoding IGF-1 receptor and different proteins that are implicated in the inflammatory process, and acute-phase responses, and consequently oxidative damage of liver begins [126], IGF-1 may play a role in HCV-associated hepatocarcinogenesis by supporting HCV infection. IGF-1 has an inhibitory effect on lipolytic enzyme lipoprotein lipase which prevents the virus cell entry in hepatoma cells [127]. Besides, there is some sort of interaction between IGF-1 and the three oncogenic HCV proteins, including, capsid protein (protein C), and two non-structural proteins NS3 and NS5A [128].

In this study, we found a significant negative correlation between hepatic expression of IGF-1 and patient age in HCC group. During childhood, there is a greater synthesis of GH-IGF-1 axis, as a result of increased production of sex steroid hormones [129]. With age the activity of this axis shows a gradual decline to protect the organism from the harmful effects of GH on age-related attenuation of DNA repair [130]. Perhaps this could be a sensible explanation of our findings. We also observed that the hepatic expression of IGF-1 was directly related to BMI of those patients. Malnutrition is a frequent consequence of chronic liver diseases which stems from inadequate dietary intake, disturbed metabolism and malabsorption [131, 132]. It has been described that malnutrition could change the GH-IGF-1 pathway by

producing GH resistance, inhibiting hepatic expression of GHR, and IGF-1 mRNA, as well as precipitating the disintegration, and reducing the bioactivity of IGF-1 [133]. However, the precise mechanism(s) remains elusive [134]. The decline of hepatic expression of IGF-1, may justify the inverse correlation between hepatic expression of IGF-1 on the one hand and some biochemical markers directly related to the severity of hepatic dysfunction; such as ALT, AST and PT on the other hand in the current study. This has promoted investigators to use the serum level of IGF-1 as an alternative to the subjective variables (ascites, hepatic encephalopathy) in Child-Pugh scoring system, for evaluating the hepatic functional reserve in HCC staging scores [135]. Also, The negative correlation between hepatic expression of IGF-1 and serum creatinine, signifies the extent of dysfunction of liver and not the kidneys, considering that chronic renal diseases usually leads to increased levels of IGF-1 [136]. Concerning the relation between hepatic expression of IGF-1 protein and the clinico-pathological characteristics of tumor, herein we observed an inverse correlation between hepatic expression of IGF-1 and vascular invasion in HCC patients. Furthermore, there was a significant association between decreased hepatic expression of IGF-1 and increased incidence of PVT in those patients. Similar finding was reported by Ikeda et al. [111]. During HCC development, increased hepatic expression of IGF-2 has been observed leaving these tissue more vulnerable to the mitogenic effects of IGF-1 [137]. IGF-1 could activate STAT5 signaling, which leads to promotion of EMT of HCC cells via downregulation of E-cadherin and upregulation of N-cadherin and vimentin [85]. Herein, the expression level of IGF-1was negatively related to Snail-1 expression, however, it was positively correlated to TGFBR2 expression.

Another valuable finding of this study concerns the increased hepatic expression of the studied proteins in a variety of cells which belong to the HCC microenvironment such as KCs, HSCs, LSECs, cholangiocytes, and stromal inflammatory cells in HCC patients. KCs are liver-resident phagocytes which play a pivotal role in different signaling pathways mediating inflammation and tumor progression [138]. Activated HSCs secret various cytokines in addition to hepatocyte growth factor that results in attenuated antitumor immunity and triggers hepatocarcinogenesis [139, 140]. LSECs contribute to occurrence of chronic liver injury and thus tumorigenesis by allowing persistence of chronic viral infections, exacerbation of fibrosis, acquisition of angiogenesis and EMT [141]. Cholangiocytes lead to liver fibrosis and hepatocarcinogesis through triggering EMT [142], and an inflammatory cytokine, Il17a/f1 that activates its receptor and thus ERK dependent pathway [143]. The main stromal inflammatory cells in the HCC microenvironment are HSCs, fibroblasts, endothelial cells, adipocytes, and immune cells; which include CD8[+] T cells, regulatory T cells, dendritic cells, and macrophages. Interactions between these cells and HCC initiate a media suitable for tumor progression [140]. Our results showed that crosstalk between GHR/STAT5/TGF1 signaling pathway and EMT inducers is required for HCC development.

Undoubtedly the current study has some limitations. First was the relatively small number of sample size. Second, the study design was retrospective. Third, STAT5 tyrosine phosphorylation status, and the functional role of each STAT5 isoform were not evaluated due to limited resources. Finally we could not rule out the role of HCV infection in downregulation of hepatic expression of GHR and its downstream pathway among HCV-associated HCC.

In conclusion, by using 1HC method, we found that down-regulation of GHR and its downstream pathway was correlated to the development of HCV-related HCC, that was associated with tumor aggressiveness and worse prognosis, irrespective of the functional status of liver. Being potent inducers of EMT, Snail-1 and TGFBR2 could be critical contributors. However, the ultimate utility of these results in practice warranted further validation by other large prospective, multi-center studies.

## Author Contributions

**Conceptualization:** Mona A. Abu El-Makarem.

**Data curation:** Ahmed A. Mohamed, Hisham A. Ali, Mahmoud R. Mohamed, Alaa El-Deen M. Mohamed, Ahmed M. El-Said, Alshymaa A. Hassnine, Hatem A. Hassan.

**Formal analysis:** Mahmoud G. Ameen.

**Investigation:** Mahmoud G. Ameen.

**Methodology:** Mariana F. Kamel, Mahmoud G. Ameen.

**Project administration:** Mona A. Abu El-Makarem.

**Resources:** Mahmoud R. Mohamed.

**Supervision:** Mona A. Abu El-Makarem, Ahmed A. Mohamed, Hisham A. Ali.

**Validation:** Mona A. Abu El-Makarem.

**Writing – original draft:** Mona A. Abu El-Makarem.

**Writing – review & editing:** Mona A. Abu El-Makarem.

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
