## [Decision Letter · Decision Letter 0]

9 Sep 2021

PONE-D-21-25660

Down-regulation of Hepatic Expression of GHR/STAT5/IGF-1 Signaling Pathway: Does It Play a Role in HCV-Associated Hepatocellular Carcinoma?

PLOS ONE

Dear Dr. Marakim,

Thank you for submitting your manuscript to PLOS ONE. After careful consideration, we feel that it has merit but does not fully meet PLOS ONE’s publication criteria as it currently stands. Therefore, we invite you to submit a revised version of the manuscript that addresses the points raised during the review process.  The manuscript cannot be accepted in the form but we encourage resubmission.

Please submit your revised manuscript within 120 days. If you will need more time than this to complete your revisions, please reply to this message or contact the journal office at plosone@plos.org. Please include the following items when submitting your revised manuscript:

We look forward to receiving your revised manuscript.

Kind regards,

Gianfranco D. Alpini

Academic Editor

PLOS ONE

Journal Requirements:

"NO - Include this sentence at the end of your statement: The funders had no role in study design, data collection and analysis, decision to publish, or preparation of the manuscript."

Reviewers' comments:

Reviewer's Responses to Questions

**Comments to the Author**

1. Is the manuscript technically sound, and do the data support the conclusions?

Reviewer #1: Yes

Reviewer #2: Partly

2. Has the statistical analysis been performed appropriately and rigorously? 

Reviewer #1: I Don't Know

Reviewer #2: I Don't Know

3. Have the authors made all data underlying the findings in their manuscript fully available?

Reviewer #1: No

Reviewer #2: Yes

4. Is the manuscript presented in an intelligible fashion and written in standard English?

Reviewer #1: Yes

Reviewer #2: Yes

5. Review Comments to the Author

Reviewer #1: The authors present a study at the role for GH/STAT5/IGF-1 axis in HCV related HCC. Majority of data is acquired retrospectively aside from IHC on FFPE samples from all patient groups. The authors find reduced GHR, STAT5 and IGF-1 expression in hepatic samples from patients with Cirrhosis and Cirrhosis + HCC compared to healthy controls. The authors recognize the limits of their study, however, consideration of the following items should be considered for publication of this work in PloS ONE literature body.

1. Title makes this seem as a review due to the question. It may benefit the authors to succinctly summarize their findings in the title as is common practice in original manuscripts.

2. Comparison of GHR/STAT5/IGF-1 in non-HCV HCC or early HCC (no cirrhosis) will lend strength to the authors findings and conclusions.

3. Images would benefit from presentation at 10X or 20X with a 200X zoom in box so that tumor/non-tumor staining is clearly visualized.

4. Due to the advanced nature of liver disease in patients, and the contradicting findings with other HCC studies on GHR/IGF-1/STAT5, the authors may consider staining for EMT and senescence (cell cycle arrest proteins) within the samples used in this study. This may shed light on the HCV-induced cirrhosis and HCC phenotype found in this study.

5. The authors may want to assess if HSC, KC and cholangiocytes are positive for GHR/IGF-1/STAT5/JAK2 signaling in HCC and cirrhosis since they participate in the tumor microenvironment.

6. The authors should confirm if all results met criteria for tukey post hoc test in ANOVA analysis. Generally, not all data have equal variance so there may be other appropriate post hoc tests.

Reviewer #2: In this study, El-Makarem and kamel investigated the GHR/STAT5/IGF-1 signaling pathway in normal, cirrhosis and HCC patient liver sections. A total of 40 patients with HCV-associated HCC were recruited in the study and were compared to 40 patients with HCV related cirrhosis without HCC, and 20 healthy controls, by immunohistochemistry analysis for the expression of GHR, STAT5 and IGF1.

The study was limited by: 1) the relative small patient sample size; 2) the investigation almost entirely relied on IHC staining method, which can be subjective because of the observers’ experience and the specificities of the antibodies. In addition, the phosphorylation and nuclear translocation are the key for Stat5 activation and transcriptional regulation of downstream genes, the author should also check the pY-Stat5 level in the HCC and control patients.

6. PLOS authors have the option to publish the peer review history of their article (what does this mean?). If published, this will include your full peer review and any attached files.

Reviewer #1: **Yes: **vik meadows

Reviewer #2: No

---

## [Author Response · Author response to Decision Letter 0]

17 May 2022

Dear Sir

Thank you for your email. We appreciate your review and comments. We had made a review of our manuscript and addressed the issues that came up in your email. The replays for the referee's questions and comments are:

Reviewer #1:

1- Title makes this seem as a review due to the question. It may benefit the authors to succinctly summarize their findings in the title as is common practice in original manuscripts.

• Title has been corrected according to your advice.

2- Comparison of GHR/STAT5/IGF-1 in non-HCV HCC or early HCC (no cirrhosis) will lend strength to the authors findings and conclusions.

• Unfortunately, inclusion of any of these two groups has been impeded by financial obstacles and the rarity of these disease entities. The recession that now plagues all countries, in particular those of the third world, has resulted in shortage of scientific research budget. For this reason, scientific researches in our institution are highly self-supporting. 

HCC is a serious health problem in Egypt and its increasing incidence is related mainly to chronic HCV infection (El-Serag HB, Kanwal F. Epidemiology of hepatocellular carcinoma in the United States: where are we? Where do we go? Hepatology 2014; 60: 1767-1775). Globally, the highest prevalence of HCV infection is in Egypt (Blach S, Zeuzem S, Manns M, et al. Global prevalence and genotype distribution of HCV infection in 2015: a modeling study. Lancet Gastroenterol Hepatol. 2017; 2: 161-176). 

HCC usually develops in cirrhotic livers, only 11% of HCCs have been reported to develop in patients with a non- cirrhotic livers with preserved liver function (Shin J, Yu JH, Jin YI, Lee JW. Incidence of clinical features of hepatic C virus-associated hepatocellular carcinoma patients without liver cirrhosis in hepatitis B virus-endemic area. J Liver Cancer 2021; 21(1): 34-44). Fibrolamellar carcinoma, a rare variant of HCC, also occurs without a background cirrhosis or hepatitis (Desai A, Sandhu S, Lai J-P, Sanhu DS. Hepatocellular carcinoma in non-cirrhotic liver: A comprehensive review. WJ Hepatol 2019; 11(1):1-18). It has been reported that HCC patients with non-cirrhotic HCV infection usually have other risk factors for hepatocarcinogenesis (Nash KL, Woodall T, Brown SM, et al. Hepatocellular carcinoma in patients with chronic hepatitis C virus infection without cirrhosis. WJ Gastroenterol. 2016; 16(32): 4061-4065).

3- Images would benefit from presentation at 10X or 20X with a 200X zoom in box so that tumor/non-tumor staining is clearly visualized. 

• This is beyond the scope of this study. The utilization of the tissue adjacent to the tumor as a control in cancer studies has been debated since Slaughter et al. (1953), first described the field "cancerization" theory, suggesting a cumulative process of carcinogenesis in which genetic alterations are acquired leaving this adjacent tissue in an intermediate, preneoplastic state (Slaughter DP, Southwick HW, Smejkal W. Field cancerization in oral stratified squamous epithelium: clinical implications of multicentric origin. Cancer 1953; 6: 963-968). This concept has been reinforced by a recent study (Aran D, Camarda R, Odegaard J, et al. Comprehensive analysis of normal adjacent to tumor transcriptomes. Nature Communications 2017; 8: 1077). That is why we were keen to compare the tumor tissue with cirrhotic tissue away from the tumor.

4- Due to the advanced nature of liver disease in patients, and the contradicting findings with other HCC studies on GHR/IGF-1/STAT5, the authors may consider staining for EMT and senescence (cell cycle arrest proteins) within the samples used in this study. This may shed light on the HCV-induced cirrhosis and HCC phenotype found in this study.

• Taken into consideration your valuable advice, we investigated the hepatic expression of Snail-1 and TGFBR2 proteins as potent inducers of EMT in the same samples used in this study (c.f., Tables 2,3 and Figure 2). Studying other molecules is beyond our financial resources and may lead to a long winded article.

5- The authors may want to assess if HSC, KC and cholangiocytes are positive for GHR/IGF-1/STAT5/JAK2 signaling in HCC and cirrhosis since they participate in the tumor microenvironment.

• Examples of expression of the studied proteins (GHR, STAT5, IGF-1, Snail-1, and TGFBR2) in a variety of stromal cells of HCC microenvironment are shown in Figure 3. The roles of these cells in the development of HCC are mentioned in the second paragraph on page 28 of the revised manuscript.

6- The authors should confirm if all results met criteria for tukey post hoc test in ANOVA analysis. Generally, not all data have equal variance so there may be other appropriate post hoc tests.

• The data were re-evaluated by using ANOVA analysis followed by Bonferroni post-hoc test that is more appropriate in this case (Bland JM, Altman DG: Multiple significance tests: The Bonferroni method. BMJ 1995; 310:170) (c.f., Table 1).

Reviewer #2:

1- The relative small patient sample size

• To obtain a power of 99%, a sample size of 40 patients with HCV-HCC was considered in this study. It was calculated at the level of 0.05 significance using G power 3.19.2 Software, that was mentioned in the subjects and methods section on page 6, first paragraph, third line of the revised manuscript.

2- The investigation almost entirely relied on IHC staining method, which can be subjective because of the observers’ experience and the specificities of the antibodies. In addition, the phosphorylation and nuclear translocation are the key for Stat5 activation and transcriptional regulation of downstream genes, the author should also check the pY-Stat5 level in the HCC and control patients.

• IHC is an important method for pathologists as it specifically visualized distribution and amount of certain molecule in the tissue. The characteristic feature that makes IHC stand out among many other laboratory tests is that it is performed without destruction of histologic architecture, so the assessment of an expression pattern of a given molecule is possible in the context of microenvironment. Moreover, the target molecule and its subcellular, cellular, and intercellular distribution could be evaluated (Schacht V, Kern JS. Basics of immunohistochemistry. J Invest Dermatol 2015; 135:e30.). But IHC is a technique that requires care in each step of the procedure. For this reason, we insisted upon careful evaluation of all components involved in each step of the IHC technique and correction of all deficiencies to obtain optimal staining. In a trial to overcome the inter-observer variability in the current study, IHC stained samples were evaluated twice in different times by two experienced pathologists, blinded for the clinco-pathological data of the study subjects. 

Checking of the level of STAT5 rather than p-STAT5 in the study groups was one of the limitations of this study that was already mentioned by authors in the limitation paragraph. However, the nuclear entrance of STAT5 is mediated by different systems and pathways, irrespective of its phosphorylation status, as mentioned in the second paragraph on page 25 of the revised manuscript.

---

## [Decision Letter · Decision Letter 1]

20 Jun 2022

PONE-D-21-25660R1Down-regulation of Hepatic Expression of GHR/STAT5/IGF-1 Signaling Pathway Fosters Development and Aggressiveness of HCV-Related Hepatocellular Carcinoma: Crosstalk with Snail-1 and Type 2 Transforming Growth Factor-beta receptorPLOS ONE

Dear Dr. Mona Abdel Rahman Abuel Makarim,

Thank you for submitting your manuscript to PLOS ONE. After careful consideration, we feel that it has merit but does not fully meet PLOS ONE’s publication criteria as it currently stands. Therefore, we invite you to submit a revised version of the manuscript that addresses the points raised during the review process.  Some of the comments previously mentioned were not addressed.  Please do so in the new resubmitted version.

We look forward to receiving your revised manuscript.

Kind regards,

Gianfranco D. Alpini

Academic Editor

PLOS ONE

Reviewers' comments:

Reviewer's Responses to Questions

**Comments to the Author**

1. If the authors have adequately addressed your comments raised in a previous round of review and you feel that this manuscript is now acceptable for publication, you may indicate that here to bypass the “Comments to the Author” section, enter your conflict of interest statement in the “Confidential to Editor” section, and submit your "Accept" recommendation.

Reviewer #1: All comments have been addressed

Reviewer #2: (No Response)

2. Is the manuscript technically sound, and do the data support the conclusions?

Reviewer #1: Partly

Reviewer #2: Partly

3. Has the statistical analysis been performed appropriately and rigorously? 

Reviewer #1: Yes

Reviewer #2: Yes

4. Have the authors made all data underlying the findings in their manuscript fully available?

Reviewer #1: No

Reviewer #2: Yes

5. Is the manuscript presented in an intelligible fashion and written in standard English?

Reviewer #1: Yes

Reviewer #2: Yes

6. Review Comments to the Author

Reviewer #1: The authors addressed comments that were within the author's original scope of the work and within their financial budgets.

Reviewer #2: This is the first revision of the original manuscript. Although the authors addressed some of issues raised by the reviewers, some other issues remained. One examples of this: the authors investigated the expression of the studied proteins (GHR, STAT5, IGF-1, Snail-1, and TGFBR2) in a variety of stromal cells of HCC microenvironment as shown in Figure 3. However, the low magnification of the images makes it extremely hard for the reviewer to examine the staining signal in those cell types. The reviewer 1 raised this issue in the comments for the previous submission, however, the authors refused to correct the problem. I strongly urge the authors to address this issue fully in the next revision.

7. PLOS authors have the option to publish the peer review history of their article (what does this mean?). If published, this will include your full peer review and any attached files.

Reviewer #1: No

Reviewer #2: No

---

## [Author Response · Author response to Decision Letter 1]

29 Jul 2022

Dear Dr. Gianfranco D. Alpini

Academic Editor

Plos One

I hope this email finds you well. First and foremost, I sincerely appreciate your positive outlook and support. The reply for the reviewer #2 comment is:

According to his valuable advice, the images are presented at 20X magnification power with a 200X zoom in box (c.f., Figure 3). I hope the images resolution will be accepted.

Kind regards,

Prof. Dr. Mona A. Abu El-Makarem

ORCID: 0000-0001-9202-4373

July 24, 2022

---

## [Decision Letter · Decision Letter 2]

10 Aug 2022

PONE-D-21-25660R2Down-regulation of Hepatic Expression of GHR/STAT5/IGF-1 Signaling Pathway Fosters Development and Aggressiveness of HCV-Related Hepatocellular Carcinoma: Crosstalk with Snail-1 and Type 2 Transforming Growth Factor-beta receptorPLOS ONE

Dear Dr. Mona Abdel Rahman Abuel Makarim,

Thank you for submitting your manuscript to PLOS ONE. After careful consideration, we feel that it has merit but does not fully meet PLOS ONE’s publication criteria as it currently stands. Therefore, we invite you to submit a revised version of the manuscript that addresses the points raised during the review process.  This is the last possibility for resubmisison. Please address one minor comment of one of the Reviewers.

Please submit your revised manuscript within 30 days. If you will need more time than this to complete your revisions, please reply to this message or contact the journal office at plosone@plos.org. Please include the following items when submitting your revised manuscript:A rebuttal letter that responds to each point raised by the academic editor and reviewer(s). You should upload this letter as a separate file labeled 'Response to Reviewers'.A marked-up copy of your manuscript that highlights changes made to the original version. You should upload this as a separate file labeled 'Revised Manuscript with Track Changes'.An unmarked version of your revised paper without tracked changes. You should upload this as a separate file labeled 'Manuscript'.If applicable, we recommend that you deposit your laboratory protocols in protocols.io to enhance the reproducibility of your results. Protocols.io assigns your protocol its own identifier (DOI) so that it can be cited independently in the future. For instructions see: https://journals.plos.org/plosone/s/submission-guidelines#loc-laboratory-protocols. Additionally, PLOS ONE offers an option for publishing peer-reviewed Lab Protocol articles, which describe protocols hosted on protocols.io. Read more information on sharing protocols at https://plos.org/protocols?utm_medium=editorial-email&utm_source=authorletters&utm_campaign=protocols.

We look forward to receiving your revised manuscript.

Kind regards,

Gianfranco D. Alpini

Academic Editor

PLOS ONE

Journal Requirements:

Reviewers' comments:

Reviewer's Responses to Questions

**Comments to the Author**

1. If the authors have adequately addressed your comments raised in a previous round of review and you feel that this manuscript is now acceptable for publication, you may indicate that here to bypass the “Comments to the Author” section, enter your conflict of interest statement in the “Confidential to Editor” section, and submit your "Accept" recommendation.

Reviewer #2: (No Response)

2. Is the manuscript technically sound, and do the data support the conclusions?

Reviewer #2: Yes

3. Has the statistical analysis been performed appropriately and rigorously? 

Reviewer #2: Yes

4. Have the authors made all data underlying the findings in their manuscript fully available?

Reviewer #2: No

5. Is the manuscript presented in an intelligible fashion and written in standard English?

Reviewer #2: Yes

6. Review Comments to the Author

Reviewer #2: Although the authors provided some 200x zoom-in box within the figure 3 (a-n), some of the image quality is not satisfying. For example, the images in zoom-in box of Figure 3-a and figure3-e is very fuzzy. I suggest the authors to provide the improved images. It was a very simple request for the last revision, I don't understand why the authors couldn't just retake some of the images and meet the publication standards.

7. PLOS authors have the option to publish the peer review history of their article (what does this mean?). If published, this will include your full peer review and any attached files.

Reviewer #2: No

---

## [Author Response · Author response to Decision Letter 2]

17 Oct 2022

Dear Dr. Gianfranco D. Alpini

Academic Editor

Plos One

I trust this email finds you well. The reply for journal requirements is:

I would like to assure you that the reference list becomes complete, correct and does not contain any retracted reference.

Note: 

In references no.: 4, 14, 52, 53, 54, 64, 79, 84, 102, 108, 123, 125, 128, 131, 136 and 144, only the front page numbering is available according to the publication guidelines. 

The reply for the reviewer #2 comment is:

I sincerely apologize for any inconvenience caused. All slides have been retaken to get clearer pictures, and this is the most possible for the efficiency of the devices available in our institute. 

Kind regards,

Prof. Dr. Mona A. Abu El-Makarem

ORCID: 0000-0001-9202-4373

26-9- 2022

---

## [Decision Letter · Decision Letter 3]

25 Oct 2022

Down-regulation of Hepatic Expression of GHR/STAT5/IGF-1 Signaling Pathway Fosters Development and Aggressiveness of HCV-Related Hepatocellular Carcinoma: Crosstalk with Snail-1 and Type 2 Transforming Growth Factor-beta receptor

PONE-D-21-25660R3

Dear Dr. Mona Abdel Rahman Abuel Makarim,

We’re pleased to inform you that your manuscript has been judged scientifically suitable for publication and will be formally accepted for publication once it meets all outstanding technical requirements.

Kind regards,

Gianfranco D. Alpini

Academic Editor

PLOS ONE

Additional Editor Comments (optional):

Reviewers' comments:

Reviewer's Responses to Questions

**Comments to the Author**

1. If the authors have adequately addressed your comments raised in a previous round of review and you feel that this manuscript is now acceptable for publication, you may indicate that here to bypass the “Comments to the Author” section, enter your conflict of interest statement in the “Confidential to Editor” section, and submit your "Accept" recommendation.

Reviewer #2: All comments have been addressed

2. Is the manuscript technically sound, and do the data support the conclusions?

Reviewer #2: Partly

3. Has the statistical analysis been performed appropriately and rigorously? 

Reviewer #2: Yes

4. Have the authors made all data underlying the findings in their manuscript fully available?

Reviewer #2: Yes

5. Is the manuscript presented in an intelligible fashion and written in standard English?

Reviewer #2: Yes

6. Review Comments to the Author

Reviewer #2: The authors have largely addressed my concerns for the image quality in the previous submissions, and I believe it is now suitable for publishing in "PLOS One".

7. PLOS authors have the option to publish the peer review history of their article (what does this mean?). If published, this will include your full peer review and any attached files.

Reviewer #2: No

---

## [Editor Report · Acceptance letter]

2 Nov 2022

PONE-D-21-25660R3 

Down-regulation of Hepatic Expression of GHR/STAT5/IGF-1 Signaling Pathway Fosters Development and Aggressiveness of HCV-Related Hepatocellular Carcinoma: Crosstalk with Snail-1 and Type 2 Transforming Growth Factor-beta receptor 

Dear Dr. Abuel Makarim:

I'm pleased to inform you that your manuscript has been deemed suitable for publication in PLOS ONE. Congratulations! Your manuscript is now with our production department. 

Kind regards, 

on behalf of

Dr. Gianfranco D. Alpini 

Academic Editor

PLOS ONE